# *HSD3B1* (c.1100C) Genotype Is Associated with Distinct Tumoral and Clinical Outcomes in Breast and Endometrial Cancers

**DOI:** 10.3390/ijms26125720

**Published:** 2025-06-14

**Authors:** Nikitha Vobugari, Allison Makovec, Samuel Kellen, Shayan S. Nazari, Andrew Elliott, Devin Schmeck, Aiden Deacon, Gabriella von Dohlen, Emily John, Pedro C. Barata, Neeraj Agarwal, Melissa A. Geller, Britt K. Erickson, George Sledge, Julie H. Ostrander, Rana R. McKay, Charles J. Ryan, Nima Sharifi, Emmanuel S. Antonarakis, Justin Hwang

**Affiliations:** 1Department of Medicine, University of Minnesota-Twin Cities, Minneapolis, MN 55455, USA; vobug002@umn.edu (N.V.); makov016@umn.edu (A.M.); kell2658@umn.edu (S.K.); schme154@umn.edu (D.S.); adeacon@umn.edu (A.D.); vondo025@umn.edu (G.v.D.); joh21009@umn.edu (E.J.); hans1354@umn.edu (J.H.O.); 2Masonic Cancer Center, University of Minnesota, Minneapolis, MN 55455, USA; gelle005@umn.edu (M.A.G.); bkeric@umn.edu (B.K.E.); 3Caris Life Sciences, Phoenix, AZ 85040, USA; snazari@carisls.com (S.S.N.); aelliott@carisls.com (A.E.); gsledge@carisls.com (G.S.); 4Department of Internal Medicine, University Hospitals Seidman Cancer Center Case Western Reserve University, Cleveland, OH 44106, USA; pedro.barata@uhhospitals.org; 5Division of Medical Oncology, Huntsman Cancer Institute (NCI-CCC), University of Utah, Salt Lake City, UT 84112, USA; neeraj.agarwal@hci.utah.edu; 6Department of Obstetrics and Gynecology, University of Minnesota-Twin Cities, Minneapolis, MN 55455, USA; 7Department of Medicine and Urology, UC San Diego School of Medicine, La Jolla, CA 92093, USA; rmckay@health.ucsd.edu; 8Memorial Sloan Kettering Cancer Center, New York, NY 10065, USA; ryanc8@mskcc.org; 9Desai Sethi Urology Institute and Sylvester Comprehensive Cancer Center, University of Miami Miller School of Medicine, Miami, FL 33136, USA; nimasharifi@miami.edu

**Keywords:** breast cancer, endometrial cancer, *HSD3B1*, genetics, genomics, transcriptomics

## Abstract

*HSD3B1* encodes an enzyme that catalyzes the conversion of adrenal precursors into potent sex steroids. A common germline variant (c.1100C) enhances this effect and is linked to breast cancer (BC) progression. As the *HSD3B1* genotypes contribute to differences in local and adrenal steroid production, their transcriptional and phenotypic effects on cancers influenced by hormonal signaling such as BC and endometrial cancer (EC)—particularly in relation to menopausal status—remain unclear. We analyzed BC and EC sequenced from patients that received diagnostic tests in oncology clinics, and we determined the germline *HSD3B1* c.1100 genotype (AA, AC, CC) from tumor DNA sequencing by using variant allele frequency, with inferred menopausal status assumed by age at molecular profiling. Whole-transcriptome RNA sequencing and gene set enrichment analysis showed that adrenal-permissive homozygous (CC) tumors in premenopausal ER + BC were enriched for hormone-related pathways, including Estrogen Response Early (NES ≈ +1.8). In premenopausal triple-negative BC, adrenal-restrictive homozygous (AA) tumors exhibited the elevated expression of immune and epithelial genes and the increased prevalence of *MED12* alterations (AA 0.25% vs. CC 8%, *p* < 0.01). In endometrioid EC, CC tumors demonstrated the suppression of immune and proliferative pathways. Postmenopausal cases had higher progesterone receptor IHC positivity (AA 75% vs. CC 83%, *p* < 0.05) and numerically more frequent *ESR1* copy number gains (AA 2.0% vs. CC 4.0%). Results highlight context-specific associations between germline *HSD3B1* genotypes and tumor biology in BC and EC.

## 1. Introduction

The subsets of breast and endometrial cancers express the estrogen receptor (ER) and can be driven by receptor signaling activity. Therefore, studying genes and pathways that regulate estrogen or progesterone receptor-associated pathways may provide information regarding the mechanisms of disease pathogenesis. The majority of these malignancies fall under the estrogen-driven category, where carcinogenesis is primarily driven by pathways influenced by estrogen and progesterone receptors signaling. In premenopausal women, the primary source of estrogen is derived from ovarian production, with a smaller contribution from extragonadal sources. In postmenopausal women, the predominant source of estrogen shifts to extragonadal conversion, through the peripheral conversion of adrenal-derived androgens into estrone and estradiol by aromatase, mainly in adipose tissue. Recent evidence has implicated the enzyme 3β-hydroxysteroid dehydrogenase-1 (3βHSD1), encoded by the hydroxy-delta-5-steroid dehydrogenase, 3 beta- and steroid delta-isomerase 1 gene (*HSD3B1*), in the synthesis of androgens from adrenal precursors [1]. Circulating dehydroepiandrosterone (DHEA) is a steroid primarily produced by the adrenal glands. 3βHSD1 acts as a rate-limiting step for the conversion of DHEA into androgens and then estrogens [2]. A germline polymorphism in the *HSD3B1* gene (c.1100C) yields two protein products of 3βHSD1. The adrenal-restrictive *HSD3B1* genotype (AA) remains a rate-limiting step for sex hormone production. In contrast, the adrenal permissive *HSD3B1* genotype (CC) yields a stabilized protein product [3], which in turn enhances this DHEA conversion to bolster the local levels of potent androgens and estrogens [4]. Importantly, in multiple clinical studies of prostate cancer patients, the *HSD3B1* CC genotype predicts poor responses to hormone therapies and increased prostate cancer-specific mortality [2,5,6].

Estrogen signaling through the ER promotes the growth and progression of subsets of breast and endometrial cancers [7]. As the ER signaling is a primary therapeutic axis, the processes that regulate estrogen production and the ligand activation of the ER also have important implications in disease progression. In hormone receptor-positive breast cancers (BC) and endometrial cancers (EC), premenopausal and postmenopausal women exhibit distinct hormonal and biological environments that may influence clinical outcomes and treatment responses. One key distinction is the shift of the major estrogen source following menopause. The *HSD3B1* genotype, which regulates adrenal-derived androgen production and its subsequent peripheral conversion to estrogen, may have particular relevance in postmenopausal settings. Related to this study, the adrenal-permissive genotype (CC) of *HSD3B1* has been preliminarily associated with poor outcomes in hormone receptor-positive BC and EC. For example, ER-positive (ER+) BC harboring the CC genotype has been shown to exhibit a greater propensity for distant metastases in postmenopausal women compared to those with the AA genotype [8]. These hormonal distinctions underscore the biological rationale for stratifying analyses by menopausal status and highlight the need to explore genotype-driven differences. Conversely, recent studies have shown that BCs and ECs harboring the adrenal-restrictive AA *HSD3B1* genotype are instead enriched in basal-like BC, as well as ECs that harbor high copy number amplifications (CNAs) and/or non-endometrioid subtypes [9]. While these findings support a role for germline *HSD3B1* at the population level, the functional consequences of germline *HSD3B1* genotypes in human tumors remain largely uncharacterized in the literature, with limited mechanistic insights into tumor-specific effects.

Our hypothesis is that *HSD3B1* genotypes could correlate with distinct tumor characteristics and clinical outcomes in pre- and postmenopausal BC and EC subtypes. Further, these genotypes may be differentially relevant in the setting of pre- and postmenopausal cancers, in which patients may have distinct sources and levels of circulating hormones. To comprehensively evaluate the role of *HSD3B1* genotypes, our study integrates germline genotype inference with transcriptomic and genomic profiling to analyze its influence on tumor biology and clinical outcomes in BC and EC, stratified by inferred menopausal status and cancer subtype.

## 2. Results

In this study, we classified samples based on the *HSD3B1* genotype as either adrenal-permissive or -restrictive, using our recently developed approach to infer germline *HSD3B1* status using the variant allele frequency (VAF) obtained from tumor DNA sequencing [10]. Distinct from prior studies, our access to paired whole exomes and transcriptomes of these samples permitted the examination of the interactions of germline genotypes with the somatic tumoral features, serving as the largest cohort used to study *HSD3B1* in BC and EC to date. Our study demonstrates that *HSD3B1* genotypes distinctly impact tumor transcriptional profiles, genomic alterations, and the expression of immune and non-immune pathways across BC and EC subtypes, in both pre- and postmenopausal tumors.

### 2.1. HSD3B1 Genotypes and Demographics Data

We examined tumoral characteristics and clinical outcomes among pre- and postmenopausal patients with BC (*n* = 1628 and 3457, respectively), stratified into ER+/HER2 negative (HER2-) (*n* = 3333) and TNBC (*n* = 1752) subtypes across *HSD3B1* genotypes. Similarly, for EC, we analyzed pre- and postmenopausal patients (*n* = 771 and 5000, respectively), further categorized into endometrioid (*n* = 3303) and non-endometrioid subtypes (serous, clear cell, and carcinosarcoma; *n* = 2468). Patient demographics and clinical characteristics stratified by the *HSD3B1* genotype and menopausal status for BC and EC are summarized in Table 1 and Table 2, respectively. The prevalence of the CC genotype ranged from 7.5% to 9.4% across BC subtypes based on the menopausal status. In EC, the CC genotype prevalence varied from 3.45% to 6.11% across subtypes, though interpretation is limited by small sample sizes in non-endometrioid EC.

### 2.2. HSD3B1 Variants Have Limited Impact on Survival Outcomes in BC and EC

We evaluated the overall survival (OS) across ER + BC, triple-negative BC (TNBC), and endometrioid EC, stratified by the *HSD3B1* genotype (AA vs. AC vs. CC) and inferred menopausal status. The median OS (in months) was compared using the Kaplan–Meier analysis. Survival curves for non-endometrioid EC could not be generated due to an insufficient sample size.

The comparisons of OS based on tumors stratified by *HSD3B1* genotypes were not statistically significant in any of the analyses (Figure 1). In ER + BC, the median OS ranged from 30.5 to 96.2 months, with survival rates appearing similar across pre- and postmenopausal patients. Among the premenopausal cohort, the CC and AA survival curves trended lower than AC up to approximately 75–100 months, after which curves overlapped. However, interpretation beyond this point is limited by small sample sizes (single-digit patient numbers). Notably, the wide confidence interval for the CC group (30.5–96.2 months) reflects variability and uncertainty, likely due to the smaller sample size (*n* = 86 for CC). In postmenopausal women with the CC genotype, our data demonstrated overlapping survival curves and confidence intervals, suggesting no conclusive survival differences between *HSD3B1* genotypes in this cohort (*p* > 0.05) (Figure 1). This may reflect the tumor heterogeneity and the advanced/metastatic nature of the ER + BC population represented in the dataset, when the molecular testing is typically pursued. In TNBC, the median OS ranged from 13.3 to 38.1 months. Survival rates were relatively lower in premenopausal patients compared to postmenopausal women. Although emerging evidence suggests that the AA genotype may be associated with poorer prognosis in hormone-independent BC [9], no statistically significant differences in OS were observed across *HSD3B1* genotypes in our TNBC cohort (Figure 1).

As a sufficient number of each *HSD3B1* genotype was only found in the 3303 ECs annotated as the endometrioid subtype (Table 2), the subsequent analyses of ECs focused on examining these tumors. In endometrioid ECs, there was a non-significant trend toward the worse median OS in premenopausal patients with the CC genotype compared to those with AC or AA genotypes (Figure 1). Among postmenopausal patients, no statistically significant survival differences were noted between genotypes. Overall, in this analysis across ER + BC, TNBC, and endometrioid EC, we found no statistically significant association between *HSD3B1* genotypes and OS. These findings suggest that additional clinical or molecular factors may influence survival outcomes in patients that received multiple lines of therapies.

### 2.3. HSD3B1 Genotypes Exhibit Different Distributions by Race

When examining patients by self-reported race, White patients had the highest proportion of the adrenal-permissive CC genotype across cancer types: approximately 10% in ER + BC, 11% in TNBC, and 7% in endometrioid EC (Figure 2A). Prior studies using aggregate global allele frequencies from dbGaP have indicated that, in a non-cancerous general population, the CC genotype frequencies are approximately 10%, 1%, and 1% for Whites, African Americans, and Asian Americans, respectively [11]. We observed CC genotype frequencies among White patients (7–11.3%) mirrored the non-cancerous general population-based expectations (Figure 2A). In contrast, CC genotype frequencies among Black and Asian/PI patients were more variable, particularly when further stratified by inferred menopausal status (Figure 2B,C). Most strikingly, the CC genotype was observed in 14.6% of Asian/PI premenopausal ER+ breast cancer patients (Figure 2B), higher than the frequency among overall Asian cohort (5.9%) (see Appendix A). This trend was also observed for premenopausal Asian patients in TNBCs (6.1% vs. 2.1%). Furthermore, significant differences in the *HSD3B1* genotype distribution were found when comparing Asian/PI and White patients in premenopausal ER + BC (*p* = 0.039) and postmenopausal TNBC (*p* = 0.001) (See Appendix A). These findings suggest a potential association between genotype variation and tumor predisposition or progression among Asian/PI patients. However, similar enrichment was not observed in Asian postmenopausal ER + BC or EC (Figure 2C). Some results may be skewed by the small sample size, particularly among Asian participants (Table 1), and warrant confirmation in future studies. Overall, these racial differences in the *HSD3B1* genotype distribution may have important clinical implications, where 3βHSD1-mediated adrenal androgen metabolism could influence tumor biology, disease progression, and treatment outcomes.

### 2.4. HSD3B1 Variants Are Associated with Somatic Alterations in BC and EC

To assess whether the *HSD3B1* c.1100 genotype is associated with specific somatic alterations, we compared the frequency of key genomic mutations, CNAs, and gene fusions between AA and CC across BC and EC, stratified by inferred menopausal status. Tumors with the heterozygous AC genotype were excluded to focus on genotype extremes (i.e., to compare the two homozygous groups). In ER + BC, significant differences in genomic alterations were observed across *HSD3B1* homozygous genotypes in each of the cancer subtypes (Figure 3 and Appendix A). Further, the differential rates of alterations were dependent on the inferred menopausal status of the patients. Among premenopausal cases, ER + BC that harbored the CC genotype exhibited the elevated rates of *CTCF* and *HOXB13* alterations (*p* < 0.05 and *p* < 0.01, respectively), as well as CNAs in *H3F3B*, *JAK3*, and *XPC* (all *p* < 0.05; ** *p* < 0.001 for *H3F3B* and *XPC*) (Figure 3A and Appendix A). In postmenopausal ER + BC, the CC genotype more frequently harbored alterations in *APC* (*p* < 0.01), CNAs in *FGFR3*, *FOXA1*, and *MAP2K2*, and fusion events involving *PDCDC1* (the gene encoding PD-1) and *BRAF* (all *p* < 0.05 to ** *p* < 0.001) (Figure 3A). These findings indicate that the inferred menopausal status and the *HSD3B1* genotypes are factors that can lead to distinct somatic features, suggestive of the tumors progressing through divergent pathways.

In TNBC, genotype-associated differences were also evident. Among premenopausal tumors, CC cases showed significantly higher rates of *GNAS* and *MED12* alterations (*p* < 0.05 and *p* < 0.01), as well as *MYB* fusions (*p* < 0.01). In postmenopausal TNBC, CC tumors demonstrated increased mutations in *KEAP1*, *KRAS*, *STK11*, and *TSC2* (*p* < 0.05 to * *p* < 0.01) (Figure 3B and Appendix A). As these genes are involved in oxidative stress responses, RAS signaling, and mTOR regulation, this suggests that the CC genotype in TNBC may be associated with a more proliferative and signal-enriched tumor phenotype in postmenopausal BC samples. KEAP1, a tumor suppressor gene that promotes antitumor immunity, is frequently mutated in non-small cell lung cancer, leading to increased NRF2 activity and poorer clinical outcomes. These mutations are associated with elevated PD-L1 expression and altered responses to immune checkpoint inhibitors in NSCLC [12]. While the relevance of KEAP1 in TNBC is still evolving, emerging evidence suggests it may also be linked to poorer outcomes [13].

In endometrioid EC, somatic profiles also differed by genotype and the inferred menopausal status. The *PTEN* function is essential toward endometrioid tumorigenesis [1]. While *PTEN* alteration events were overall prevalent, in premenopausal EC, CC tumors harbored an increased frequency of *PTEN* alterations compared to AA tumors (*p* < 0.05) (Figure 3C and Appendix A). Increased PTEN alterations may have contributed to the poorer survival trends observed among the CC genotypes in the premenopausal cohort (Figure 1). In postmenopausal EC, AA tumors were enriched for mutations in *BCOR*, *JAK1*, and *ZFHX*3 (*p* < 0.05), while CC tumors had higher frequencies of *KRAS*, *MTMR2*, and *MTA2* mutations and CNAs in *ESR1*, *ARID2*, and *KLF6* (*p* < 0.05). Immunohistochemistry (IHC) measurements of PR positivity were also significantly elevated in CC tumors (*p* < 0.05), consistent with a more hormone-responsive phenotype (Figure 3C and Appendix A). These findings indicate that the *HSD3B1* genotype and inferred menopausal status together define endometrioid EC with divergent molecular features.

### 2.5. Differences in Gene Expression Across HSD3B1 Genotypes Is Observed in BC and EC

To investigate the potential tumor-intrinsic effects of the *HSD3B1* genotype, we performed differential gene expression analyses comparing BC and EC tumor subtypes that harbor either the CC or AA genotypes. Similar to our prior genomic analysis, tumors with the heterogeneous AC genotype were excluded. In ER + BC tumors, across *HSD3B1* genotypes in both pre- and postmenopausal tumors, there was minimal variation in the transcriptome, including in the expression of canonical hormone receptor genes (*ESR1*, *ESR2*, *PGR*) (Figure 4, and Appendix A). In TNBC, transcriptional differences between *HSD3B1* genotypes were overall more pronounced. In premenopausal TNBC, tumors harboring the CC genotype had reduced expression of inflammatory genes, like *S100A7* and *LY6D* [14,15] (*p*-values = 0.0006 and = 0.0052, respectively), and basal-like genes, such as *SPRR2G* [16] (*p*-value = 0.0039). Furthermore, in both pre- and postmenopausal TNBC, we noted that *SERF1A,* a gene with no known function and part of a gene signature used to estimate the likelihood of BC recurrence [17], was upregulated in the CC genotype (*p*-value = 0.0004 and 0.0083, respectively). Hormone-related genes, like *ESR2* and *NR3C2* (encoding the mineralocorticoid receptor), showed no significant differential expression between genotypes across inferred menopausal statuses in TNBC (Figure 4B and Appendix A). In endometrioid ECs, *HSD3B1* genotypes differed in expression in only a few genes. Across both menopausal statuses, there were no significant differences in canonical hormone receptor genes. In pre- but and not postmenopausal tumors, the CC genotype harbored the increased expression of *CSNS1*, in which high expression has been associated with worse survival outcomes in BC [18] and *SERPINA11*, which has been implicated in tumor suppression in hepatocellular carcinoma [19] (Figure 4C and Appendix A). Overall, transcriptional variation dependent on *HSD3B1* genotypes differed by the cancer type, with the majority of differences seen in TNBC and less variation observed in ER + BC and endometrioid EC.

### 2.6. HSD3B1 Regulates Hallmark Signaling Pathways and Immune-Related Genes

We then explored *HSD3B1* genotype-associated transcriptional programs at the pathway level by conducting gene set enrichment analysis (GSEA) [20] across BC and EC subtypes. ER + BC harboring the CC genotype exhibited the enrichment of pancreatic beta cells, the only significantly upregulated hallmark pathway. In contrast, multiple immune-related pathways, including interferon gamma response and inflammatory response, were significantly downregulated in CC tumors compared to AA, indicating a reduction in immune activation within the tumor microenvironment (Figure 5A and Appendix A). In TNBC, CC tumors—especially in premenopausal patients—showed the significant enrichment of proliferative and metabolic pathways (Figure 5A and Appendix A). These included MYC targets V1, E2F targets, G2M checkpoint, DNA repair, and MTORC1 signaling, suggesting that the CC genotype may be linked to enhanced cell cycle progression and stress-response signaling. This pattern was not accompanied by the significant enrichment of immune-related pathways, reinforcing a proliferation-dominant transcriptional phenotype.

In endometrioid EC, CC tumors exhibited the broad depletion of transcriptional programs across both inferred menopausal groups. However, in premenopausal CC tumors, the interferon alpha response was the only hallmark pathway significantly enriched, indicating selective immune engagement despite overall transcriptional suppression (Figure 5A and Appendix A). These findings suggest that the *HSD3B1* CC genotype is associated with distinct transcriptional states: selective immune suppression in ER + BC, proliferative signaling in TNBC, and widespread downregulation in EC—with limited immune compensation in premenopausal tumors.

To examine if *HSD3B1* genotypes influence the expression of individual cell surface genes that are currently drug targets or ones that are associated with immunity, we also performed the focused analysis of a collection of tumor-associated cell surface and immune markers [10] across both BC and EC subtypes and menopausal status (Figure 5B and Appendix A). Here, based on (log_2_) fold changes in expression between CC and AA tumors, while we observed trends of the up- or downregulation of these genes, none of these genes exhibited significant differences based on an FDR of < 0.05.

## 3. Discussion

While prior studies have primarily examined *HSD3B1* c.1100 genotypes in prostate cancer outcomes [2,5,6] and their impact on androgen deprivation therapy response [5], emerging evidence has extended their relevance to other hormone-driven cancers such as BC and EC. Specifically, the adrenal-permissive CC genotype has been linked to disease progression in postmenopausal ER + BC via enhanced intratumoral estrogen synthesis [21,22], while emerging evidence links AA genotypes to non-hormone-driven BC and EC [9]. These genotype-driven differences in steroid metabolism may influence tumor behavior, prognosis, and treatment response. We analyzed the clinical, genomic, and transcriptomic landscape of *HSD3B1* genotypes across BC and EC subtypes, stratified by inferred menopausal status—representing the largest cohort to date exploring these associations.

Contrary to early reports, we did not observe statistically significant differences in OS based on the *HSD3B1* genotype in any cancer subtype or menopausal group. Unlike prior studies showing worse outcomes—such as increased distant metastases and reduced survival—in postmenopausal women with the CC genotype [21], our data revealed overlapping survival curves and confidence intervals, indicating no definitive survival differences between *HSD3B1* genotypes in this cohort. Poorer non-significant survival trends were noted among CC genotypes compared to AA or AC among premenopausal EC patients. Our cohort, selected from a genomically tested population likely enriched for advanced or metastatic disease, may reflect a clinical context where the influence of adrenal-derived steroid metabolism is diminished due to tumor burden or prior therapies. In TNBC and EC, where hormonal dependence is variable, the absence of genotype-associated survival differences may similarly reflect tumor heterogeneity or the limited role of *HSD3B1* in late-stage disease. These findings suggest that, while the *HSD3B1* genotype may have prognostic relevance in the early-stage, treatment-naive settings, its role in advanced cancers may be less clear and potentially overshadowed by additional clinical and molecular drivers. Further studies in more homogeneous, early-stage populations are needed to clarify the potential clinical utility of *HSD3B1* genotyping for prognostication and/or treatment selection. An ongoing clinical trial (NCT05183828) is evaluating whether the *HSD3B1* germline variant influences tumor proliferation markers, hormone receptor expression, and letrozole response in postmenopausal women with ER + HER2- BC [23].

This study is the first to highlight racial disparities in the *HSD3B1* genotype distribution across all tumor subtypes. White patients generally exhibited the highest proportion of the homozygous adrenal-permissive CC genotype (10%), consistent with known population data, while the CC genotype is markedly less common (~1%) in Black and East Asian non-cancerous populations [11]. We observed notable differences in the *HSD3B1* genotype distribution when stratified by the cancer subtype and inferred menopausal status. Most prominently, CC genotype enrichment was seen in Asian/PI premenopausal ER + BC (14.6%) and TNBC (6.1%) subgroups, exceeding population-based expectations and suggesting a potential link to tumor predisposition or progression. Cross-race comparisons between Asian/PI and White patients revealed significant differences in the *HSD3B1* genotype distribution in premenopausal ER + BC and postmenopausal TNBC. These patterns were not observed in postmenopausal ER + BC and EC, and findings should be interpreted cautiously due to limited sample sizes. Supporting the broader relevance of this observation, our previously published prostate cancer cohort also showed CC genotype frequencies of 9–10% among Black and Asian patients [10], again supporting the potential role of *HSD3B1* in hormone-driven cancer biology and cancer predisposition. Prior studies, such as those by Kruse et al., focused exclusively on White populations and did not address racial variability [22]. Our findings raise the possibility that inherited differences in adrenal androgen metabolism may contribute to racial disparities in cancer predisposition and progression, particularly in hormonally influenced cancers. While these patterns were not directly tied to survival in our cohort and results could be skewed due to a lower sample size among non-White population specifically Asian patients in our cohort, they warrant further investigation in racially diverse, early-stage populations to better understand the interplay between the *HSD3B1* genotype, genetic ancestry, and clinical outcomes.

Our findings in which tumors harbored context-dependent signaling pathways based on the *HSD3B1* adrenal-permissive and -restrictive genotypes drives our understanding of both proliferative and metabolic activity in the clinical subtypes of BC. The role of *HSD3B1* genotypes in extragonadal androgen metabolism and steroidogenesis has been well established [2], but we present here, to our knowledge, the first consortium study that deploys the RNA-based measurements of oncogenic and immune regulatory pathways in BC and EC specimens derived from human tumors. Through the GSEA studies, we uncovered that CC tumors in premenopausal patients, particularly those with TNBC, show significant enrichment in MTORC1 signaling, MYC target gene sets, and DNA repair pathways. These findings altogether implicate that the *HSD3B1* genotypes may stratify tumors with divergent metabolic and proliferative functions. In these tumors, the enrichment of DNA repair pathways may further suggest the potential for the adrenal permissive genotype to regulate genomic stability or respond to DNA-damaging therapies, which is consistent with previous reports linking steroidogenic enzymes to DNA repair modulation. Independent of *HSD3B1* genotypes, MYC and MTORC1 are critical regulatory pathways in aggressive BC [24,25]. Therefore, it is critical to examine mechanisms related to how the permissive genotype contributes to the enhancement of these signaling pathways, and why this is unique to premenopausal TNBCs. In the case that 3βHSD1 acts as an upstream regulator for these processes, the emergence of experimental agents that target 3βHSD1 function may be worth exploring in these clinical settings [26]. It is also curious why the adrenal-permissive *HSD3B1* genotypes were associated with these pathways in TNBC patients, as one may surmise that these hormone-receptor-negative tumors may have little dependency on enzymes that regulate this process. Through the analysis of NGS data, it remains unclear if 3βHSD1 has direct or indirect functions beyond regulating estrogen production.

Unlike TNBCs, in ER + BCs, the CC genotype exhibited the depletion of immune-related hallmark pathways, which is suggestive of immune suppression. While these findings are specific to BC, prior studies on prostate cancer indicate that steroid hormone signaling can suppress antitumor immunity. In prostate cancer, androgen receptor (AR) signaling has been shown to downregulate MHC class I (HLA) expression, thereby reducing tumor immunogenicity, a process that is reversed by AR inhibition [27]. AR blockade has also been shown to promote antitumor immunity and increases in interferon signaling, consistent with the broader literature linking hormone signaling to the repression of immune pathways [28,29]. These observations suggest that hormone-driven immune suppression, which is well characterized in prostate cancer, may also play a role in the immune-depleted phenotype we observe in ER + BC. Interestingly, in postmenopausal ER+ tumors that harbor the permissive *HSD3B1* genotype, the only upregulated pathway in this context was the pancreatic beta cell signature, a pathway previously associated with both insulin signaling and metabolic stress in tumors [30,31]. This again suggests that 3βHSD1 may have non-canonical functions. Collectively, our observations in BC indicate that *HSD3B1* genotypes are potential regulators of oncogenic signaling and metabolic plasticity across BC subtypes, and this is dependent on the inferred menopause/hormonal status. Therefore, we propose that future studies should consider investigating if tumors that express the protein products of adrenal-permissive or -restrictive *HSD3B1* in fact yield specific therapeutic vulnerabilities, such as sensitivity to mTOR pathway inhibitors or agents that target tumors with dependency on DNA repair signaling. Importantly, this would support the use of stratifying BC patients by *HSD3B1* genotypes and reporting this information in the clinical setting.

In endometrioid ECs, we found that independent of the inferred menopausal status, tumors harboring the CC genotype generally exhibit reduced oncogenic signaling pathways. The consistent negative enrichment of these pathways is perhaps indicative of a transcriptionally quiescent phenotype in the CC tumors. This may be aligned with prior studies that noted reduced proliferative signaling in hormone-sensitive tumors with high PR expression [32]. In premenopausal ECs, the only pathway that was significant was the interferon alpha response pathway, which in the setting of tumor indicates an association with antitumor immunity in such tumors [33]. Additionally, in premenopausal EC, CC genotype tumors showed a higher frequency of PTEN alterations compared to AA, which may have contributed to the poorer survival trends observed in the CC group. In postmenopausal tumors, genomic analyses also revealed that adrenal-permissive patients were more likely to exhibit positivity of PR via IHC, as well as CNAs in *ESR1* and *ERBB2*. While the magnitude of differences in prevalence may be subtle, the consistency of genes associated with dysregulated hormone receptor signaling indicate a potential interaction with the *HSD3B1*-permissive genotype in postmenopausal patients. While our data do not support a universally pro-oncogenic role for the CC genotype in ECs, they instead suggest a genotype that could be associated with transcriptional quiescence and hormone receptor preservation, similar to what we find in BC subtypes, and the further mechanistic investigations of *HSD3B1*-permissive and -restrictive protein functions could contribute to novel insights in patients with EC.

For over twelve years, *HSD3B1* has been studied in prostate cancer, where the adrenal-permissive allele is associated with enhanced extragonadal and intratumoral androgen synthesis, resistance to androgen deprivation therapies, and poor outcomes [2,4]. Despite this, its functions in oncogenesis in other hormonally driven cancers were relatively unexplored, especially in human cancer specimens. Regardless of its established role in androgen regulation and androgen signaling in prostate cancer, our findings suggest that tumors harboring the adrenal-permissive *HSD3B1* exhibit profound differences in biological pathways. In prostate cancer, our recent study conducted in over 5000 tumor specimens, based on the identical method of patient stratification and using the same Caris diagnostic tests, indicated that the permissive genotype drives did not promote significant differences in the expression of the same genes or the activity of the oncogenic pathways we found in these BCs and ECs [10]. However, as the diagnostic testing of tumors is biased toward late-stage disease, the role of *HSD3B1* in the early stages of cancer remains to be determined in human tumors. Regardless, these cross-cancer differences accentuate the need for a more expansive examination with regard to the clinical relevance of *HSD3B1*, in not just BC and EC, but likely in other cancer types as well. As of now, our findings support the potential of germline *HSD3B1* genotype as a clinically relevant genetic marker that could be incorporated into the existing diagnostic workflows for these endocrine-driven cancers.

### Limitations of the Study

This study has several inherent limitations. It is retrospective in design, and detailed clinical information, such as menopausal status, tumor grade, disease stage, and treatments administered, was therefore not available. Immune cell profiling was based on bulk RNA sequencing rather than single-cell resolution, which may limit interpretability. The actual menopausal status was not available; therefore, the key conclusions derived based on the inferred annotation must be further explored through a study with the actual menopausal status at the time of cancer diagnosis. Here, the menopausal status was approximated using the age at the time of molecular profiling, in which samples from patients older than 55 years were annotated as menopausal. This is based on prior meta-analyses and population studies in BC patients [21,22,34,35]. Additionally, patient data were categorized by self-reported race, leading to the exclusion of individuals with unreported race classifications.

Among endometrioid EC cases, the ER status was determined at the level of demographic descriptive analysis (Table 2) that showed approximately 85–92% of endometrioid cases to be ER-positive across *HSD3B1* genotypes. However, endometrioid EC cases in this study were analyzed as a single group without further stratification by the ER status or histologic grade. Given that approximately 10–15% of endometrioid tumors are ER-negative and that grade 3 tumors are biologically distinct—often associated with TP53 mutations and worse outcomes—this grouping may have introduced heterogeneity. Future analyses should consider restricting comparisons to ER-positive, grade 1–2 endometrioid tumors to reflect estrogen-driven biology more accurately. Finally, the classifications of ECs in this study are based on histology; therefore, the findings may differ if these samples were re-classified based on molecular classification schemes, which are now being adapted in oncology clinics [36].

## 4. Materials and Methods

### 4.1. Patient Specimens

Formalin-fixed, paraffin-embedded (FFPE) specimens from BC and EC patients were manually microdissected to enrich tumor tissue and underwent molecular testing at Caris Life Sciences (Phoenix, AZ, USA), a College of American Pathologists/Clinical Laboratory Improvement Amendments (CAP/CLIA)-certified laboratory. Hematoxylin and eosin (H&E)-stained, formalin-fixed, paraffin-embedded (FFPE) slides of the patient’s tumor underwent review by a board-certified pathologist or trained pathologist assistant. Tumor enrichment was achieved by harvesting targeted tissue using manual microdissection techniques.

### 4.2. Patient Cohort and HSD3B1 Genotype

We included ER + BC and TNBC patients (*n* = 5085) and EC patients (*n* = 5700) queried from the dataset, including samples that had undergone NextGen Sequencing of DNA and RNA whole-transcriptome data. Patients’ subtype was stratified via variant allele frequency, and IHC detection was used to determine the ER/TNBC status in the BC cohort. HER2 positive BC cases were excluded to maintain the study’s focus on evaluating the impact of *HSD3B1* genotypes within hormone-driven (ER-active) and non-hormone-driven (TNBC/basal-like) subtypes. Inferred menopausal status was assumed based on age at the time of molecular profiling, with <55 years classified as premenopausal and ≥55 years as postmenopausal. EC cases were classified by their histological subtype—endometrioid, serous, clear cell, and carcinosarcoma—and also categorized by their ER status (see Table 2). Baseline demographic and clinical characteristics for the BC and EC cohorts—stratified by the *HSD3B1* genotype, cancer subtype, inferred menopausal status, self-reported race, tumor mutational burden (TMB), and microsatellite instability (MSI)—are summarized in Table 1 and Table 2. Endometrioid EC cases were further analyzed as a single group (includes both type I and II endometrioid EC) for survival outcomes and tumor profiling and were not further stratified by the ER status due to sample size limitations. Appendix A shows annotations used by Caris to group endometrioid EC subtypes. Based on this cohort, we focused on analyzing endometrioid ECs and excluded other EC Subtypes beyond descriptive demographic analysis due to the limited samples in the premenopausal setting (*n* = 4), which limits the statistical power for any analysis. Analysis based on the histologic grades and molecular classification of EC is not included as this is not the primary study focus. The Kaplan–Meier overall survival analysis curves were generated and are presented in Figure 1.

The germline variant of *HSD3B1* was determined from evaluating the VAF of *HSD3B1* at position c.1100. Tumors with 0% c.1100A VAF character were considered adrenal-permissive (CC), 40–60% for c.1100A were considered heterozygous (AC), and those with 100% VAF for c.1100A were considered adrenal-restrictive (AA) (Table 1 and Table 2). Any VAF outside of these ranges was considered uncategorized.

### 4.3. Whole-Exome Sequencing and NGS

NGS was performed on genomic DNA isolated from formalin-fixed, paraffin-embedded (FFPE) tumor samples using the NovaSeq 6000 and the NextSeq sequencing platforms (Illumina, Inc., San Diego, CA, USA). Caris tests are certified by Clinical Laboratory Improvement Amendments (CLIA). Samples in this study were not prospectively sequenced and were processed similar to prior studies [10]. Caris does not process samples more than five years old, which is in part due to the known effects of degradation. Internal standards are applied to evaluate RNA integrity and sequencing, and results are only reported if they pass these quality controls.

Following DNA extraction, a custom-designed SureSelect biotinylated baits panel (Agilent Technologies, Santa Clara, CA, USA) was used to enrich 592 whole-gene targets that recurrently altered in cancer. Library preparation was performed using SureSelectXT reagents, followed by normalization and pooling. Demultiplexed FASTQ files underwent bioinformatic processing for sequence alignment, variant detection, quality-based filtering, and annotation of the sequencing data.

A combination of bait panels was used for WES analysis, including the SureSelect Human All Exon v7 panel (Agilent Technologies, Santa Clara, CA, USA) covering 99% of coding regions, a hybrid pull-down panel of baits designed to enrich 720 genes recurrently altered in cancer, an intronic fusion panel, a pathogenic baits panel, and a 250K SNP backbone panel to assist with gene amplification/deletion measurements and other analyses (Agilent Technologies, Santa Clara, CA, USA). Library preparation, quantification, normalization, and pooling were performed using the Bravo Automated Liquid Handling Platform and the Agilent Tape Station 4200 (Agilent Technologies). Whole-exome sequencing was performed on the NovaSeq 6000 platform (Illumina, Inc., San Diego, CA, USA) (RRID:SCR_016387). The average sequencing depth of coverage was >500×.

Genetic variants were categorized as “pathogenic”, “likely pathogenic”, “variant of unknown significance”, “likely benign”, or “benign” according to the ACMG standards.

### 4.4. Whole-Transcriptome Sequencing

RNA was extracted using an RNA FFPE Extraction Kit (Qiagen, Hilden, Germany), and RNA quality and quantity were determined using the Agilent TapeStation (Agilent Technologies, Santa Clara, CA, USA). A SureSelect Human All Exon v7 panel of biotinylated RNA baits (Agilent Technologies) were hybridized to the synthesized and purified cDNA targets, and the bait–target complexes were amplified in a post-capture PCR reaction. The resultant libraries were quantified and normalized, and the pooled libraries were denatured and diluted. Library preparation, quantification, normalization, and pooling were performed using the Bravo Automated Liquid Handling Platform and the Agilent Tape Station 4200 (Agilent Technologies). WTS was performed on the Novaseq 6000 System (Illumina, San Diego, CA, USA) (RRID:SCR_016387).

### 4.5. Immune Signatures

The relative abundance of immune cell infiltrates in the tumor microenvironment were calculated from the WTS data using quanTIseq (RRID:SCR_022993) [37].

### 4.6. Statistical Analysis

For statistical analysis, Kruskal–Wallis, Chi-square, and Fisher’s exact tests were used. For the demographic tables, FDR *q*-values of <0.05 (Benjamini–Hochberg procedures) were considered significant. For the GSEA ranked analysis test, statistical significance was assessed with an NES score and a calculated FDR value. The GSEA analysis was performed using the GSEApy python package v1.1.6.

### 4.7. TMB and MSI/dMMR

TMB was calculated by the addition of all non-synonymous missense, nonsense, inframe insertion/deletion, and frameshift mutations found per tumor. A cutoff value of ≥10 mutations per MB was used. The MSI/MMR status was inferred using a combination of NGS and IHC markers compared to the hg19 human genome.

## 5. Conclusions

Our work presents a novel analysis of the association between inferred germline *HSD3B1* c.1100 genotypes and the somatic landscape in BC and EC. This study indicates that across the subtypes of BC and EC, in both pre- and postmenopausal tumors, *HSD3B1* genotypes distinctly influence transcriptional profiles, genomic alterations, and differences in the expression of immune and non-immune pathways. While our findings are hypothesis-generating and require further validation, the influence of *HSD3B1* genotypes in BC and EC appears substantial and warrants continued investigation for potential relevance to treatment decision making.

## Figures and Tables

**Figure 1 ijms-26-05720-f001:**
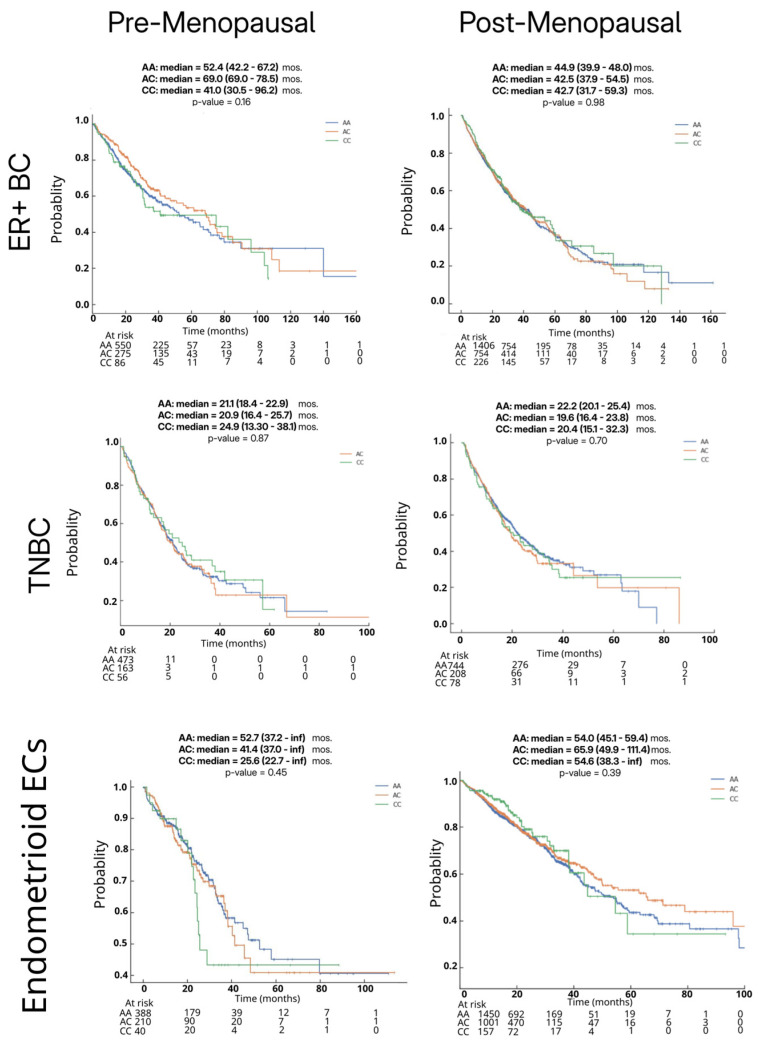
Survival outcomes by germline *HSD3B1* genotype across pre- and postmenopausal ER + BC, TNBC, and endometrioid EC patients. Kaplan–Meier overall survival curves are shown for AA, AC, and CC genotypes in each cohort. Statistical significance was assessed using Cox proportional hazards regression analysis.

**Figure 2 ijms-26-05720-f002:**
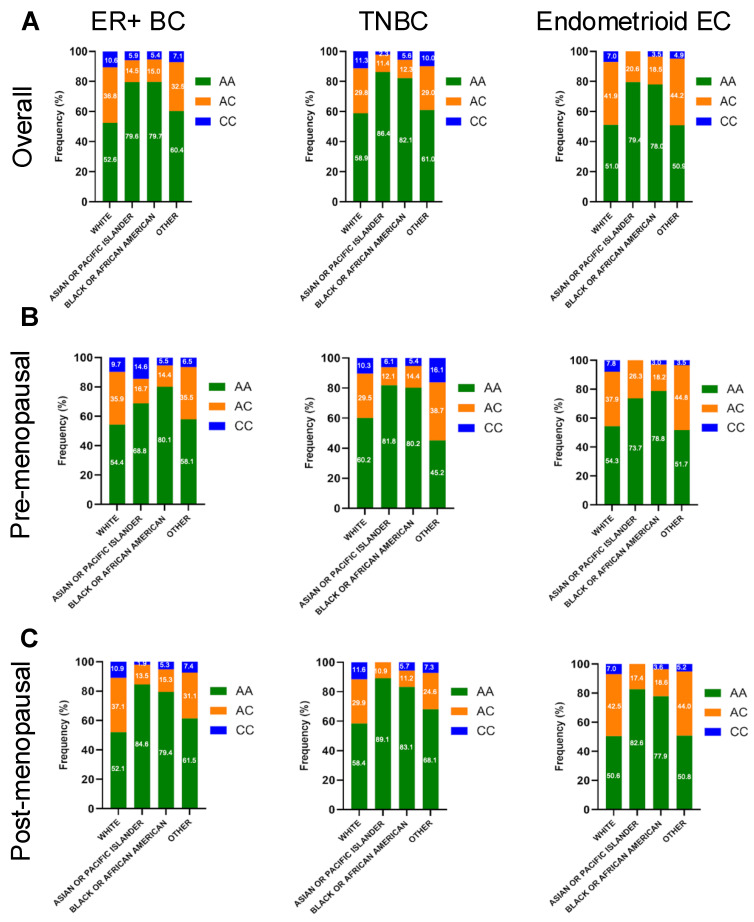
Frequency of genomic alterations in germline *HSD3B1* AA vs. CC tumors distributed by race across pre- and postmenopausal ER + BC, TNBC, and endometrioid EC cases. (**A**) Distribution across overall population, (**B**) Distribution across pre-menopausal population, (**C**) Distribution across post-menopausal population. Bar plots display percentage of tumors with specific alterations across genotype. See more details in Appendix A.

**Figure 3 ijms-26-05720-f003:**
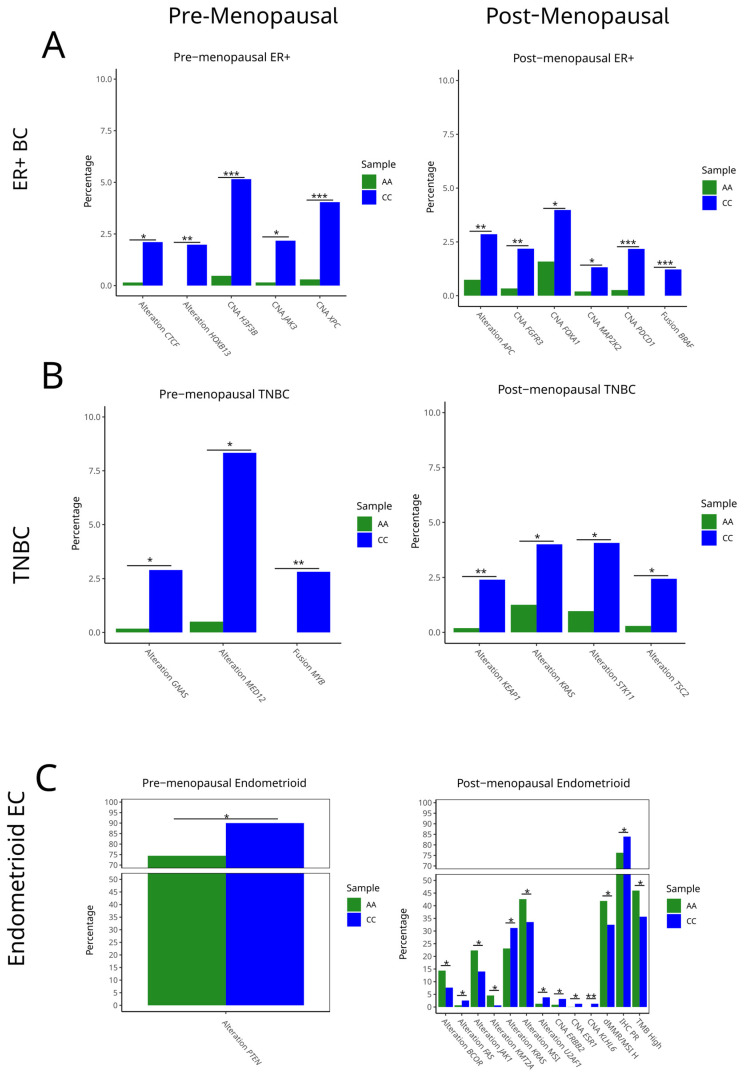
Genomic alterations by germline *HSD3B1* genotype across (**A**) pre- and post-menopausal ER + BC, (**B**) pre- and post-menopausal TNBC, and (**C**) pre- and post-menopausal endometrioid EC cases. Bar plots display frequency of select pathogenic mutations, copy number amplifications (CNAs), and gene fusions in adrenal-permissive (CC) and adrenal-restrictive (AA) tumors, stratified by the cancer type and inferred menopausal status. Tumors with the heterozygous AC genotype were excluded. Differences in alteration frequencies were evaluated using Fisher’s exact test (* *p* < 0.05, ** *p* < 0.01, *** *p* < 0.001). See Appendix A for further details.

**Figure 4 ijms-26-05720-f004:**
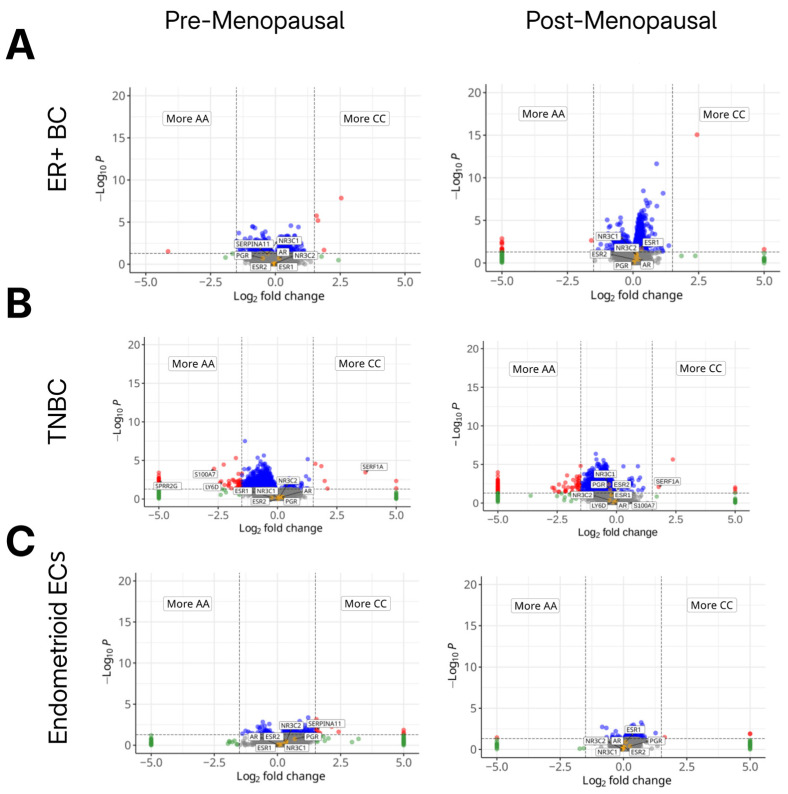
Differences in gene expression across germline *HSD3B1* genotypes. Volcano plots showing differential gene expression between adrenal-permissive (CC) and adrenal-restrictive (AA) tumors across cancer types and inferred menopausal states. (**A**) ER+ BC, (**B**) TNBC, and (**C**) endometrioid EC, each stratified by inferred menopausal status. Log_2_ fold change reflects gene expression in CC relative to AA (positive = up in CC; negative = up in AA). AC tumors were included in cohort but not directly compared in these analyses. Red and blue points indicate significantly upregulated genes in CC or AA tumors, respectively (FDR < 0.05, |log_2_FC| > 1). Orange dots reflect genes that are hormone receptors (ESR1/2, PGR, AR, NR3C1/2). Transcriptional differences were minimal in ER + BC but more pronounced in premenopausal TNBC and EC. See further details in Appendix A.

**Figure 5 ijms-26-05720-f005:**
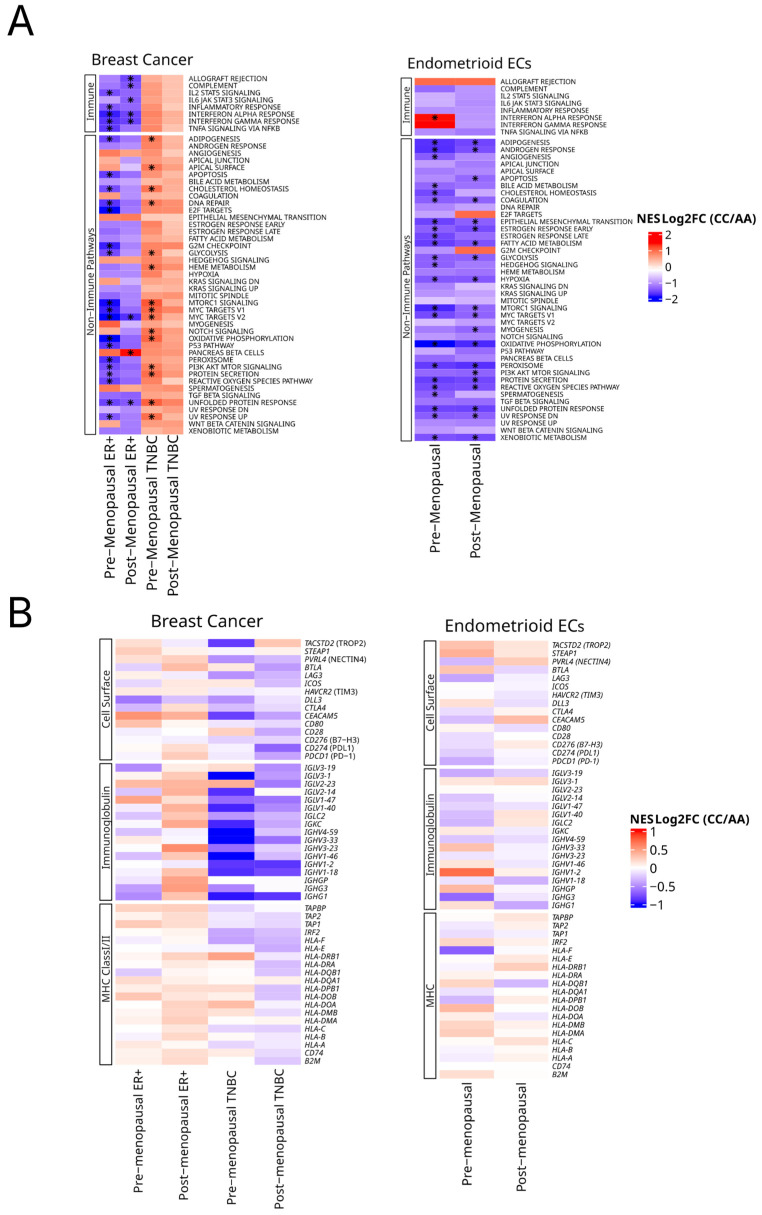
Pathway and gene expression analysis across germline *HSD3B1* genotypes. (**A**) Gene set enrichment analysis (GSEA) of 50 hallmark pathways reflecting enrichment in CC vs. AA. This is stratified by tumor type and inferred menopausal status. Hallmark pathways are grouped by immune and non-immune signaling in BC and EC. Heatmaps represent normalized enrichment scores (NES) with asterisks indicating FDR of <0.05. See Appendix A for further details. (**B**) Relative expression of cell surface, immunoglobin and MHC class I/II gene pathways as grouped by BCs and ECs.

**Table 1 ijms-26-05720-t001:** Baseline characteristics of breast cancer cohort (*n* = 5085), including *HSD3B1* genotype, race, inferred menopausal status, molecular subtype, and TMB/MSI status.

Patient Characteristics, *N* (%)	Adrenal Restrictive (AA), *N* (%)	Heterozygous (AC), *N* (%)	Adrenal Permissive (CC), *N* (%)	Total	FDR
Patients					
Genotype	3220 (63.32)	1415 (27.83)	450 (8.85)	5085 (100)	
Subtype Data					
ER+/HER2-	1980 (59.41)	1040 (31.20)	313 (9.39)	3333 (100)	
TNBC	1240 (70.78)	375 (21.40)	137 (7.82)	1752 (100)	
Inferred Menopausal Status (assumed based on age at molecular profiling)					
Premenopausal (≤55 years)	1040 (63.88)	444 (27.27)	144 (8.85)	1628 (100)	
Postmenopausal (>55 years)	2180 (63.06)	971 (28.09)	306 (8.85)	3457 (100)	
Menopausal Subtype Data					
Premenopausal ER+	560 (60.54)	279 (30.16)	86 (9.30)	925 (100)	
Premenopausal TNBC	480 (68.28)	165 (23.47)	58 (8.25)	703 (100)	
Postmenopausal ER+	1420 (58.97)	761 (31.60)	227 (9.43)	2408 (100)	
Postmenopausal TNBC	760 (72.45)	210(20.02)	79 (7.53)	1049 (100)	
Race Data (Self-Identified)					
White	1428 (54.82)	890 (34.17)	287 (11.02)	2605 (100)	3.50 × 10^−36^
Black Or African American	753 (81.32)	123 (13.28)	50 (5.40)	926 (100)	6.80 × 10^−35^
Asian Or Pacific Islander	142 (80.23)	24 (13.56)	11 (6.21)	177 (100)	1.60 × 10^−5^
Other	144 (60.50)	75 (31.51)	19 (7.98)	238 (100)	4.20 × 10^−1^
Unknown	284 (64.84)	124 (28.31)	30 (6.85)	438 (100)	3.04 × 10^−1^
Tumor Mutational Burden (TMB) and Microsatellite Instability					
TMB High	880 (56.63)	600 (38.61)	74 (4.76)	1554 (100)	1.00
dMMR/MSI-H High	771 (56.86)	520 (38.35)	65 (4.79)	1356 (100)	1.00

**Table 2 ijms-26-05720-t002:** Baseline characteristics of endometrial cancer cohort (*n* = 5771), including *HSD3B1* genotype, race, inferred menopausal status, molecular subtype, estrogen receptor status, and TMB/MSI status.

Patient Characteristics, *N* (%)	Adrenal Restrictive (AA), *N* (%)	Heterozygous (AC), *N* (%)	Adrenal Permissive (CC), *N* (%)	Total	*q*-Values (FDR)
Patients					
Genotype	3499 (60.63)	1935 (33.53)	337 (5.84)	5771 (100.00)	
Subtype Data					
Carcinosarcoma	114 (71.70)	32 (20.13)	13 (8.18)	159 (100)	
Clear Cell	153 (62.20)	76 (30.89)	17 (6.91)	246 (100)	
Endometrioid	1876 (56.80)	1230 (37.24)	197 (5.96)	3303 (100)	
Serous	1270 (65.13)	571 (29.28)	109 (5.59)	1950 (100)	
Inferred Menopausal Status (assumed based on age at molecular profiling)					
Premenopausal (≤55 years)	485 (62.91)	242 (31.39)	44 (5.71)	771 (100)	
Postmenopausal (>55 years)	3014 (60.28)	1693 (33.86)	293 (5.86)	5000 (100)	
Menopausal Subtype Data					
Premenopausal Endometrioid	399 (60.92)	216 (32.98)	40 (6.11)	655 (100)	
Postmenopausal Endometrioid	1477 (55.78)	1014 (38.29)	157 (5.93)	2648 (100)	
Premenopausal Other Subtypes	86 (74.14)	26 (22.41)	4 (3.45)	116 (100)	
Postmenopausal Other Subtypes	1537 (65.35)	679 (28.87)	136 (5.78)	2352 (100)	
ER-Positive Prevalence by Subtype					
Premenopausal Endometrioid	345 (60.85)	185 (32.63)	37 (6.52)	567 (100)	1.00
Postmenopausal Endometrioid	1270 (55.12)	892 (38.72)	142 (6.16)	2304 (100)	1.00
Premenopausal Other Subtypes	42 (72.41)	14 (24.14)	2 (3.45)	58 (100)	9.5 × 10^−1^
Postmenopausal Other Subtypes	849 (65.36)	387 (29.79)	63 (4.85)	1299 (100)	9.0 x 10^−1^
Race Data (Self-Identified)					
White	1597 (52.67)	1205 (39.74)	230 (7.59)	3032 (100)	1.80 × 10^−1^
Black or African American	882 (79.89)	199 (18.03)	22 (2.08)	1104 (100)	1.13 × 10^−37^
Asian or Pacific Islander	146 (78.49)	38 (20.43)	2 (1.08)	186 (100)	3.17 × 10^−18^
Other	127 (53.14)	93 (38.91)	19 (7.95)	239 (100)	2.88 × 10^−6^
Unknown	238 (56.53)	159 (37.77)	24 (5.70)	421 (100)	5.92 × 10^−2^
Tumor Mutational Burden (TMB) and Microsatellite Instability					
TMB High	880 (56.63)	600 (38.61)	74 (4.76)	1554 (100)	1.00
dMMR/MSI-H High	771 (56.86)	520 (38.35)	65 (4.79)	1356 (100)	1.00

## Data Availability

This study did not generate any new raw DNA/RNA sequencing data but rather used the existing data from the database through a formal letter of intent and a subsequent data use agreement. Here, we researched the de-identified data collected in a real-world health care setting, and this is subject to controlled access for privacy and proprietary reasons. When possible, derived data supporting the findings of this study have been made available within the paper and its Appendix A. Other data can be acquired through a letter of intent to Caris Life Sciences (https://www.carislifesciences.com/letter-of-intent/ accessed on 12 June 2025). Additional inquiries can be sent to Andrew Elliott at aelliott@carisls.com.

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
