# Peer review of "HSD3B1 (c.1100C) Genotype Is Associated with Distinct Tumoral and Clinical Outcomes in Breast and Endometrial Cancers"

_ijms, 2025, doi:10.3390/ijms26125720_

Round 1
Reviewer 1 Report
Comments and Suggestions for Authors
The manuscript entitled “HSD3B1 (c.1100C) genotype is associated with distinct tumoral and clinical outcomes in breast and endometrial cancers” focuses on the impact of a common germline variant within HSD3B1 gene on tumoral transcriptional profile, cancer phenotype and outcomes in BC and EC. The topic of the manuscript is within the scope of the journal, would be interesting to the readership, especially to the clinicians and researchers interested in molecular basis of common female hormone-driven cancers. Despite the limitations recognized by the authors, the study is well designed, includes a large study population and provides a valuable insight into a role of c.1100C variant in BC and EC molecular pathogenesis.
The manuscript is relatively well written, well-structured and fluently presented. Material and methods presented the required details, while the presentation of the Results is informative, illustrative and the major conclusions are supported by the findings. There are merely some minor corrections needed:
- Line 26-28: Sentence in the Abstract section should be rephrased.
- Line 57: remove parentheses.
- Lines 59-60: A is a variant, AA is a genotype. Mind the terminology.
- The Results section should include a descriptive analysis of the data presented within Table 1. This section starts immediately with OS analysis, without any reference to genotype frequencies, distribution across subtypes, etc.
- Lines 130-133: reference to previous findings should be placed in the Discussion section, not in Results.
Author Response
For Research Article Response to Reviewer 1 Comments:
- Summary: We thank the reviewer for taking the time to review this manuscript and offering thoughtful feedback. Please find the detailed responses below and the corresponding revisions/ corrections highlighted in the re-submitted files.
|
2. Questions for General Evaluation |
Reviewer’s Evaluation |
Response and Revisions |
|
Does the introduction provide sufficient background and include all relevant references? |
Yes |
We appreciate that the reviewer found our paper sufficient and well supported. |
|
Are all the cited references relevant to the research? |
Yes |
|
|
Is the research design appropriate? |
Yes |
|
|
Are the methods adequately described? |
Yes |
|
|
Are the results clearly presented? |
Yes |
|
|
Are the conclusions supported by the results? |
Yes |
|
|
|
||
The manuscript entitled “HSD3B1 (c.1100C) genotype is associated with distinct tumoral and clinical outcomes in breast and endometrial cancers” focuses on the impact of a common germline variant within HSD3B1 gene on tumoral transcriptional profile, cancer phenotype and outcomes in BC and EC. The topic of the manuscript is within the scope of the journal, would be interesting to the readership, especially to the clinicians and researchers interested in molecular basis of common female hormone-driven cancers. Despite the limitations recognized by the authors, the study is well designed, includes a large study population and provides a valuable insight into a role of c.1100C variant in BC and EC molecular pathogenesis. The manuscript is relatively well written, well-structured and fluently presented. Material and methods presented the required details, while the presentation of the Results is informative, illustrative and the major conclusions are supported by the findings.
Thank you for your thoughtful and encouraging feedback; we greatly appreciate your recognition of the study’s design and relevance to the field.
- Point-by-point response to Comments and Suggestions for Authors
Minor corrections suggested by Reviewer 1:
Comment 1: Line 26-28: Sentence in the Abstract section should be rephrased.
Thank you for pointing this out. This has been now rephrased to “As the HSD3B1 genotypes contributes to differences in local and adrenal steroid production, their transcriptional and phenotypic effects on cancers influenced by hormonal signaling such as BC and endometrial cancer (EC)—particularly in relation to menopausal status—remains unclear.”
Comment 2: Line 57: remove parentheses.
Thank you for looking into this typographical error. This has been corrected.
Comment 3: Lines 59-60: A is a variant, AA is a genotype. Mind the terminology.
We appreciate you for pointing this out. The term “variant” has been appropriately updated to “genotype” in this sentence. (Revised manuscript line 61)
Comment 4: The Results section should include a descriptive analysis of the data presented within Table 1. This section starts immediately with OS analysis, without any reference to genotype frequencies, distribution across subtypes, etc.
Thank you for this feedback. We have now added a paragraph in the Results section highlighting the descriptive analysis of the introductory demographic data, including the information presented in Table 1.
“2.1. HSD3B1 Genotypes and Demographics Data: We examined tumoral characteristics and clinical outcomes among pre- and postmenopausal patients with BC (n = 1,628 and 3,457, respectively), stratified into ER+/HER2- (n = 3,333) and TNBC (n = 1,752) subtypes across HSD3B1 genotypes. Similarly, for EC, we analyzed pre- and postmenopausal patients (n = 771 and 5,000, respectively), further categorized into endometrioid; n = 3,303 and non-endometrioid subtypes (serous, clear cell, and carcinosarcoma; n = 2,468). Patient demographics and clinical characteristics stratified by HSD3B1 genotype and menopausal status for BC and EC are summarized in Tables 1 and 2, respectively. The prevalence of the CC genotype ranged from 7.5% to 9.4% across BC subtypes by menopausal status. In EC, CC genotype prevalence varied from 3.45% to 6.11% across subtypes, though interpretation is limited by small sample sizes in non-endometrioid EC. “
Comment 5: Lines 130-133: reference to previous findings should be placed in the Discussion section, not in Results.
Thank you for this suggestion. We have placed the sentence from Results to Discussion section—
“Unlike prior studies showing worse outcomes —such as increased distant metastases and reduced survival—in postmenopausal women with the CC genotype, our data revealed overlapping survival curves and confidence intervals, indicating no definitive survival differences between HSD3B1 genotypes in this cohort.” This is now referenced in the Discussion section and Removed from Results section accordingly.
Reviewer 2 Report
Comments and Suggestions for Authors
- Please rephrase: Given that HSD3B1 amplifies local steroid production, its transcriptional and phenotypic effects may differ across hormonal states, its influence across tumor contexts remains unclear.
- We analyzed BC and endometrial cancer (EC) tumors - Cancers or tumors?
- Breast and endometrial cancers are classified into estrogen-dependent and independent subtypes. - On what basis? Because this is not the pathological, neither the oncological classification.
- The Introduction contains too much information regarding oesotrogen, while the true mechanism in these tumors are not detailed.
- In this study, we classified samples based on HSD3B1 genotype as either adrenal permissive or restrictive, using our recently developed approach to infer germline HSD3B1 status using the variant allele frequency (VAF) obtained from tumor DNA sequencing[11]. We examined tumoral characteristics and clinical outcomes among pre- and postmenopausal patients with breast cancer (BC) (n = 1,628 and 3,457, respectively), strat-
ified into hormone-dependent (n = 3,333) and hormone-independent (n = 1,752) subtypes. Similarly, for endometrial cancer (EC), we analyzed pre- and postmenopausal patients (n= 771 and 5,000, respectively), further categorized into hormone-dependent (endometrioid; n = 3,303) and hormone-independent subtypes (serous, clear cell, and carcinosarcoma; n = 2,468). Distinct from prior studies, our access to paired whole exomes and transcriptomes of these samples permitted the interrogation of the interactions of germline variants with the somatic tumoral features, serving as the largest cohort of tumors used to study HSD3B1 in BC and EC to date. Our study demonstrates that HSD3B1 genotypes distinctly impact tumor transcriptional profiles, genomic alterations, and the expression of immune and non-immune pathways across BC and EC subtypes, in both pre- and post-menopausal tumors. - This should belong to the Results section. - How were ER+ BCs identified? Via immunohistochemistry?
- Endometrioid cancer has various molecular subtypes. Which ones were included in this study?
- Title for Table 1 is rather unusual.
- We evaluated overall survival (OS) across ER+ BC, triple-negative BC (TNBC), and en-109
dometrioid EC, stratified by HSD3B1 genotype (AA vs. AC vs. CC) and menopausal status. Patient demographics and clinical characteristics stratified by HSD3B1 genotype and men-
opausal status for BC and EC are summarized in Tables 1 and 2, respectively. - This sentence states endometrioid carcinomas were investigated, then in Table, multiple other subtypes were also included. - No comparisons reached statistical significance across any group (Figure 1). - Please rephrase. It is not obvious what this sentence tries to explain.
- ER positive BC patients: were this luminal A or B patients? Because prognosis may differ between them, as well.
- There are no p values in the section that explains the OS data. Same applies to the following section, for example, race, etc.
- Abbreviations are not in alphabetic order.
- Figure 1: Aren't these Kaplan-Meier curves? However, it is not stated in the Materials section.
- Differential Gene Expression across HSD3B1 variants is observed in BC and EC - Differential?
- The literature data are barely summarized in the Discussion.
- Unlike TNBCs, in ER+ BCs, the CC genotype exhibited depletion of immune-related
hallmark pathways, which is suggestive of immune suppression. - Does this mean that ER+ BC suppresses the immune system more than TNBC? - RNA sequencing should not be performed in paraffin embedded blocks older than a year. Therefore, how did you manage to do that? Or were the tissues older?
- Actual menopausal status was not available - However, the study was relying on this information a lot!
- Endometrioid EC cases in this study were analyzed as a single group without further
stratification by ER status or histologic grade. - It is more problematic, that other tumors were also included, that are not endometrioid carcinomas, or you are misusing the terms endometrioid and endometrial. A pathologist really should be included in this study, because there are major flaws regarding the classification of tumors.
Author Response
For Research Article Response to Reviewer 2 Comments:
- Summary: We thank the reviewer for taking the time to review this manuscript and offering thoughtful feedback. Please find the detailed responses below and the corresponding revisions/ corrections highlighted in the re-submitted files. These insightful comments have significantly improved the scope of our manuscript. We remain open to incorporating any additional suggestions.
|
2. Questions for General Evaluation |
Reviewer’s Evaluation |
Response and Revisions |
|
Does the introduction provide sufficient background and include all relevant references? |
Must be improved |
We appreciate the reviewer’s feedback and have revised the manuscript accordingly, incorporating the comments as detailed below. |
|
Are all the cited references relevant to the research? |
Must be improved |
Additional citations for the methods have been added. |
|
Is the research design appropriate? |
Must be improved |
Further demographic details on endometrioid EC have now been clarified, as outlined below. |
|
Are the methods adequately described? |
Must be improved |
Additional methodological details have been incorporated in response to the feedback. |
|
Are the results clearly presented? |
Must be improved |
Changes have been incorporated as addressed in below comments. |
|
Are the conclusions supported by the results? |
Must be improved |
Changes have been incorporated as addressed in below comments. |
|
Quality of English Language (x) The English could be improved to more clearly express the research |
||
|
|
||
- Point-by-point response to Comments and Suggestions for Authors (In red font)
Comment 1: Please rephrase: Given that HSD3B1 amplifies local steroid production, its transcriptional and phenotypic effects may differ across hormonal states, its influence across tumor contexts remains unclear.
Thank you for the comment and we agreed this could be worded more clearly. Therefore, this has been now rephrased to:
“ As the HSD3B1 genotypes contributes to differences in local and adrenal steroid production, their transcriptional and phenotypic effects on cancers influenced by hormonal signaling such as BC and endometrial cancer (EC)—particularly in relation to menopausal status—remains unclear.”
Comment 2: We analyzed BC and endometrial cancer (EC) tumors - Cancers or tumors?
Thank you for pointing out this typographical error in the Abstract. We have removed the word tumor after abbreviations BC and EC.
This sentence is now edited to: “We analyzed BC and EC, sequenced from real-world patients and we inferred germline HSD3B1c.1100 genotype………”
Comment 3: Breast and endometrial cancers are classified into estrogen-dependent and independent subtypes. - On what basis? Because this is not the pathological, neither the oncological classification.
We appreciate the reviewer’s thoughtful comment.
We were interested in evaluating the impact of HSD3B1 genotypes within known clinically actionable BC and EC subtypes, which include hormone-driven (ER-active) and non-hormone-driven (TNBC/basal-like) subtypes. These definitions meet the clinical criteria for treatment decision making in oncology clinics for the use of hormone VS non-hormone therapies and reflects our intended groups to study HSD3B1 genotypes.
To the reviewer’s point, we recognize that estrogen “dependency” is more of a functional readout that requires substantial evaluation of the tumors, as some TNBCs may not be entirely independent, whereas as some ER+ BCs are not dependent of ER. We agree these terminologies do not accurately describe the cohort, and have therefore excluded the word “dependence” or “independence” throughout the study, such as
--Introduction (line 46-49)- edited to “Breast and endometrial cancers are classified based on their association with estrogen signaling. The majority……… where carcinogenesis is primarily driven by pathways influenced by estrogen and progesterone receptors signaling.”
-- Results subsection 2.1; (line 111-116) “We examined tumoral characteristics and clinical outcomes among pre- and postmenopausal patients with BC (n = 1,628 and 3,457, respectively), stratified into ER+/HER2- (n = 3,333) and TNBC (n = 1,752) subtypes…………. further categorized into endometrioid; n = 3,303 and non-endometrioid subtypes (serous, clear cell, and carcinosarcoma; n = 2,468)”
We have also clarified our stratification approach in the Materials and Methods section under 4.2: Patient Cohort and HSD3B1Genotype, as:
In breast cancer, “The ER-active group included patients who met both the luminal A/B classification by PAM50 and the ER-positive/HER2-negative criteria by IHC. The ER-inactive group included patients who met both the basal classification by PAM50 and the ER-negative/HER2-negative criteria by IHC. This classification approach aligns more closely with the study’s mechanistic emphasis on ER signaling.”
In endometrial cancer, we have now included ER status based on IHC testing in the demographics section and Table 2, showing that approximately 85–92% of endometrioid cases were ER-positive."
Materials and Methods subsection 4.2 is now edited to;
“EC cases were classified by histological subtype—endometrioid, serous, clear cell, and carcinosarcoma—and categorized by ER status. Baseline demographic” ………. “Endometrioid EC cases were analyzed as a single group and were not further stratified by ER status due to sample size limitations. Supplementary Table 8 shows annotations used by Caris to group endometrioid EC subtypes. Based on this cohort, we focused on analyzing endometroid ECs and excluded other EC Subtypes beyond descriptive demographic analysis due to the limited samples in the pre-menopausal setting (n=4), which limits the statistical power for any analysis. Analysis based on histologic grades and molecular classification of EC is not included as this is not the primary study focus.”
Comment 4: The Introduction contains too much information regarding oesotrogen, while the true mechanism in these tumors are not detailed.
We appreciate this insightful comment, and similar to the prior 2 comments, we recognize that the function and mechanism of HSD3B1 has not been characterized in human tumors. As much of the work is done in at the cell lines, animal models, or at the population level, and the role of HSD3B1 has not been truly tested in human tumors, as most of the prior functional work is done in cell lines or observational at the patient level. The reviewer will recognize that measuring estrogen activity in human tumors is a challenge and there is no clinical application to do so. Because we cannot truly test this in tumors, our language in the revised manuscript has largely focused on the role of estrogens in BC and ECs, rather than a functional readout for HSD3B1 function.
Given the role of HSD3B1 genotypes in modulating adrenal androgen synthesis and their prognostic significance in prostate cancer, we investigated their impact on tumor features and clinical outcomes in hormone-driven female cancers—specifically breast and endometrial—while accounting for menopausal status due to the shift in estrogen source. Prior studies1,2 have linked the CC genotype to poorer outcomes in postmenopausal breast cancer. To guide readers through our rationale, we have incorporated this hypothesis in the Introduction, emphasizing the role of estrogen variability by menopausal status. We agree with the reviewer’s comment and that further studies are warranted to further elucidate the underlying mechanisms.
We have added this sentence to the Introduction section:
“While these findings support a role for germline HSD3B1 at the population level, the functional consequences of germline HSD3B1genotypes in human tumors remain largely uncharacterized in literature, with limited mechanistic insight into tumor-specific effects.”
Comment 5: In this study, we classified samples based on HSD3B1 genotype as either adrenal permissive or restrictive, using our recently developed approach to infer germline HSD3B1 status using the variant allele frequency (VAF) obtained from tumor DNA sequencing[11]. We examined tumoral characteristics and clinical outcomes among pre- and postmenopausal patients with breast cancer (BC) (n = 1,628 and 3,457, respectively), stratified into hormone-dependent (n = 3,333) and hormone-independent (n = 1,752) subtypes. Similarly, for endometrial cancer (EC), we analyzed pre- and postmenopausal patients (n= 771 and 5,000, respectively), further categorized into hormone-dependent (endometrioid; n = 3,303) and hormone-independent subtypes (serous, clear cell, and carcinosarcoma; n = 2,468). Distinct from prior studies, our access to paired whole exomes and transcriptomes of these samples permitted the interrogation of the interactions of germline variants with the somatic tumoral features, serving as the largest cohort of tumors used to study HSD3B1 in BC and EC to date. Our study demonstrates that HSD3B1 genotypes distinctly impact tumor transcriptional profiles, genomic alterations, and the expression of immune and non-immune pathways across BC and EC subtypes, in both pre- and post-menopausal tumors. - This should belong to the Results section.
Thank you for this feedback. We have now move this to a new paragraph in the Results section highlighting the descriptive analysis of the introductory demographic data, including the information presented in Table 1 and updated as below.
“2.1. HSD3B1 Genotypes and Demographics Data:
We examined tumoral characteristics and clinical outcomes among pre- and postmenopausal patients with BC (n = 1,628 and 3,457, respectively), stratified into ER+/HER2- (n = 3,333) and TNBC (n = 1,752) subtypes across HSD3B1 genotypes. Similarly, for EC, we analyzed pre- and postmenopausal patients (n = 771 and 5,000, respectively), further categorized into endometrioid; n = 3,303 and non-endometrioid subtypes (serous, clear cell, and carcinosarcoma; n = 2,468). Patient demographics and clinical characteristics stratified by HSD3B1 genotype and menopausal status for BC and EC are summarized in Tables 1 and 2, respectively. The prevalence of the CC genotype ranged from 7.5% to 9.4% across BC subtypes by menopausal status. In EC, CC genotype prevalence varied from 3.45% to 6.11% across subtypes, though interpretation is limited by small sample sizes in non-endometrioid EC. “
Comment 6: How were ER+ BCs identified? Via immunohistochemistry?
Comment 11: ER positive BC patients: were this luminal A or B patients? Because prognosis may differ between them, as well.
Thank you for the insightful questions and the opportunity to provide further clarification. Related to both Comments 6 and 11, our classification was of ER+ was based on an agreement of IHC and transcriptional profiling by PAM50 (Luminal A/B). Both criteria must be met prior to inclusion into this study.
The information on classification approach is stated under subsection “Patient Cohort and HSD3B1 Genotype” under Materials and Methods section as below:
“The ER-active group included patients who met both the luminal A/B classification by PAM50 and the ER-positive/HER2-negative criteria by IHC. The ER-inactive group included patients who met both the basal classification by PAM50 and the ER-negative/HER2-negative criteria by IHC. This classification approach aligns more closely with the study’s mechanistic emphasis on ER signaling.”
It is certainly plausible that survival outcomes in ER+ breast cancer include both luminal A and B subtypes. We chose to group luminal A and B tumors together, which are also ER+ by IHC, in comparison to TNBC/basal tumors that are ER−, to define two extreme comparator cohorts, as the main study focus is on ER signaling and HSD3B1 genotypes.
Comment 7: Endometrioid cancer has various molecular subtypes. Which ones were included in this study?
Comment 9: We evaluated overall survival (OS) across ER+ BC, triple-negative BC (TNBC), and endometrioid EC, stratified by HSD3B1 genotype (AA vs. AC vs. CC) and menopausal status. Patient demographics and clinical characteristics stratified by HSD3B1 genotype and menopausal status for BC and EC are summarized in Tables 1 and 2, respectively. - This sentence states endometrioid carcinomas were investigated, then in Table, multiple other subtypes were also included.
Comment 20: Endometrioid EC cases in this study were analyzed as a single group without further
stratification by ER status or histologic grade. - It is more problematic, that other tumors were also included, that are not endometrioid carcinomas, or you are misusing the terms endometrioid and endometrial. A pathologist really should be included in this study, because there are major flaws regarding the classification of tumors.
Thank you for pointing out these classification questions and for the opportunity to clarify these points.
Related to these three Reviewer comments, we did not group the endometrial cancer subtypes together in any analysis. We understand that clear cell, carcinosarcomas, and serous are distinct. The reason these were not further analyzed is because of the limited power. As pointed out in Table 2, when considering these 3 subtypes (labeled “Other Subtypes”), there were only 4 samples (3.45%) in the pre-menopausal setting. Any downstream analysis and interpretation would therefore be based on limited statistical power and therefore the conclusions may not be significant.
We understand this was not clear in the original manuscript and may have caused confusion, and therefore specifically stated in the revised manuscript that “based on this cohort, we focused on analyzing endometroid ECs and excluded other EC Subtypes beyond descriptive demographic analysis due to the limited samples in the pre-menopausal setting (n=4), which limits the statistical power for any analysis.”
This has been updated in Materials and Methods section addressing above comments-
“Endometrioid EC cases were further analyzed as a single group (includes both type I and II endometrioid EC) for survival outcomes and tumor profiling and were not further stratified by ER status due to sample size limitations. Supplementary Table 8 shows annotations used by Caris to group endometrioid EC subtypes. Based on this cohort, we focused on analyzing endometroid ECs and excluded other EC Subtypes beyond descriptive demographic analysis due to the limited samples in the pre-menopausal setting (n=4), which limits the statistical power for any analysis. Analysis based on histologic grades and molecular classification of EC is not included as this is not the primary study focus.”
As addressed in comment 3, we have now included ER status based on IHC testing in the demographics section and Table 2, showing that approximately 85–92% of endometrioid endometrial cancer cases were ER-positive.
This has been updated in Limitations section as well- “Among endometrioid EC cases, ER status was determined at level of demographic descriptive analysis (Table 2), that showed approximately 85–92% of endometrioid cases were ER-positive across HSD3B1 genotypes. However, endometrioid EC cases in this study were analyzed as a single group without further stratification by ER status or histologic grade…”
In addition, we realized error in labelling figure 3C- now edited from “pre- menopausal endometrial” to “pre-menopausal endometrioid” and “post-menopausal endometrial” to “post-menopausal endometrioid.”
Comment 8: Title for Table 1 is rather unusual.
Thank you for the feedback. We have now modified the titles of Table 1 and 2, respectively as below.
“Table 1: Baseline Characteristics of the Breast Cancer Cohort (n = 5085), Including HSD3B1 Genotype, Race, Menopausal Status, Molecular Subtype, and TMB/MSI Status”
“Table 2: Baseline Characteristics of the Endometrial Cancer Cohort (n = 5771), Including HSD3B1 Genotype, Race, Menopausal Status, Molecular Subtype, Estrogen Receptor Status, and TMB/MSI Status”
Comment 10: No comparisons reached statistical significance across any group (Figure 1). - Please rephrase. It is not obvious what this sentence tries to explain.
Thank you for the suggestion. In the Results, this has been edited to: “Comparisons of OS based on tumors stratified by HSD3B1 genotypes were not statistically significant in any of the analyses (Figure 1).”
Comment 12: There are no p values in the section that explains the OS data. Same applies to the following section, for example, race, etc.
P values are included in Figure 1. We have now explicitly stated this in the Results section for clarity, as reflected in the revised text addressing Comment 10.
“Comparisons of OS based on tumors stratified by HSD3B1 genotypes were not statistically significant in any of the analyses (Figure 1).”
Comment 13: Abbreviations are not in alphabetic order.
Thank you for pointing this out. This was originally based on the order in which the abbreviation was used through the study.
We have now made appropriate changes in the Abbreviations section.
Comment 14: Figure 1: Aren't these Kaplan-Meier curves? However, it is not stated in the Materials section.
Figure 1 legend and Materials and Methods section has been appropriately modified to state this.
“Figure 1: Survival outcomes by germline HSD3B1 genotype across pre- and postmenopausal ER+ BC, TNBC, and endometrioid patients. Kaplan Meier overall survival curves are shown for AA, AC, and CC genotypes in each cohort. Statistical significance was assessed using Cox proportional hazards regression analysis.”
“Kaplan-Meier overall survival analysis curves were generated and are presented in Figure 1.”
Comment 15: Differential Gene Expression across HSD3B1 variants is observed in BC and EC - Differential?
We have corrected the language here to better reflect our intended statement for subsection 2.5 title:
“Differences in gene expression based on HSD3B1 genotypes in both BC and EC".
Comment 16: The literature data are barely summarized in the Discussion.
We appreciate the reviewer’s comment.
In the revised manuscript, we have integrated relevant prior studies throughout the manuscript, providing detailed citations and contextualizing our findings within the existing body of knowledge. Our discussion emphasizes how our results compare and contrast with prior reports, noting concordant and discordant findings, and we outline potential biological mechanisms informed by previous work1–3. Given the complexity and emerging nature of this field, we have carefully referenced the most pertinent literature to support our interpretations while focusing on novel insights from our large, multi-ethnic cohort. To our knowledge, this is the largest study to date evaluating germline HSD3B1 genotype in relation to transcriptomic and genomic tumor characteristics derived from patient tumor samples in breast and endometrial cancers. While our study represents the largest and diverse cohort to date, we have carefully acknowledged its limitations and emphasized that findings may not be generalizable beyond the included population.
We have also included following relevant addition with citations to our revised MS Discussion and Limitations section:
- An ongoing clinical trial (NCT05183828) is evaluating whether the HSD3B1 germline variant influences tumor proliferation markers, hormone receptor expression, and letrozole response in postmenopausal women with ER+ HER2- BC.4
- Additional population based studies that used age >55 as cut off for menopause5,6.
Comment 17: Unlike TNBCs, in ER+ BCs, the CC genotype exhibited depletion of immune-related
hallmark pathways, which is suggestive of immune suppression. - Does this mean that ER+ BC suppresses the immune system more than TNBC?
No, this does not. As indicated in the key for Figure 5A, all comparisons are performed to evaluate differences between tumors that harborer the CC and AA genotypes, that is the central question throughout the manuscript. To clarify, the analysis and depiction reflect immune pathway activity when comparing tumors with the CC against the AA HSD3B1 genotypes. This indicates that of ER+ BCs, CC is immune suppressive relative to AA, whereas in TNBCs, this was not the case.
To the reviewer's point, if one wishes to determine the differences in immune activity between ER+ BCs and TNBCs, one must directly compare these pathways in all of the samples annotated as ER+ BCs against all of the TNBCs through GSEA. We did not conduct this revision as this is not the central narrative of the study.
Comment 18: RNA sequencing should not be performed in paraffin embedded blocks older than a year. Therefore, how did you manage to do that? Or were the tissues older?
Thank you for your insightful question.
Almost all samples presented in this study have been sequenced in the diagnostic testing, which requires rapid processing, and this is generally done within 7-days.
We agree with the reviewer that FFPE has traditionally been a concern for NGS platforms with low read depth (<50X). Based on these historical technologies, a few misread sequences could lead to misleading findings7. The NGS technology from Caris targets 1,500X coverage, which would eliminate single or limited reads that could be misinterpreted as variants and would impact mapping to references8. Generally, the combination of using shorter reads and significantly higher coverage has allowed NGS technology to be applied onto FFPE in the diagnostic setting. Relevant to the concern, Caris tests are certified by Clinical Laboratory Improvement Amendments (CLIA), which ensures the quality of the diagnostic testing and that reporting must be benchmarked.
Regarding the 2nd concern, as mentioned, Caris typically processes diagnostic tests within 1 week of receiving the sample. While Caris does accept research samples, they do not accept samples that are more than five years old, which is in part due to the known effects of degradation. Any samples that are older than five years would have to be approved by the executive medical director and internal pathology team prior to sequencing. Importantly, any results that have been reported back to an ordering physician and subsequently added to database used for research, such as that found in CODEai, would have to pass quality control metrics, else the results would be reported as “Quality not sufficient” and no RNA expression values would be reported. Thus, while some samples included in the study may be more than one year old at the time of sequencing, most will be less than one year old, and all samples would have met the quality metrics necessary for reporting the final results.
The following statement has been added to the Materials and Methods to clarify the approach:
“Caris tests are certified by Clinical Laboratory Improvement Amendments (CLIA). Samples in this study were not prospectively sequenced, and were processed similar to prior studies [10]. Caris does not process samples more than five years old, which is in part due to the known effects of degradation. Internal standards are applied to evaluate RNA integrity and sequencing, results are only reported if they pass these quality controls.“
Comment 19: Actual menopausal status was not available - However, the study was relying on this information a lot!
Thank you for pointing this out.
We have acknowledged this as a key limitation of the study and have also included it in the abstract and figure legend to ensure accurate interpretability for readers. The reviewer may appreciate that in large cohort studies, menopausal status may not be documented, and this is not cataloged in the Caris POA database, which largely focused on cancer diagnostic testing data and clinical outcomes. In light of this, we used age as a proxy for menopausal status, applying age cutoffs as recommended by prior meta-analyses in epidemiologic studies in breast cancer.
We updated Limitation section as follows:
“Actual menopausal status was not available. Menopausal status was approximated using the age at the time of molecular profiling, in which samples from patients older than 55 years were annotated as menopausal. This is based on prior meta-analyses and population studies in BC patients [21, 22, 34, 35].”
References:
- Flanagan MR, Doody DR, Voutsinas J, et al. Association of HSD3B1 Genotype and Clinical Outcomes in Postmenopausal Estrogen-Receptor-Positive Breast Cancer. Ann Surg Oncol. 2022;29(11):7194-7201. doi:10.1245/s10434-022-12088-w
- Kruse ML, Patel M, McManus J, et al. Adrenal-permissive HSD3B1 genetic inheritance and risk of estrogen-driven postmenopausal breast cancer. JCI Insight. 2021;6(20):e150403. doi:10.1172/jci.insight.150403
- McManus JM, Vargas R, Bazeley PS, Schumacher FR, Sharifi N. Association Between Adrenal-Restrictive HSD3B1 Inheritance and Hormone-Independent Subtypes of Endometrial and Breast Cancer. JNCI Cancer Spectrum. 2022;6(5):pkac061. doi:10.1093/jncics/pkac061
- University of Washington. Association of HSD3B1 Genotype With Response to Preoperative Letrozole Therapy Among Postmenopausal Women With Estrogen-Receptor Positive (ER+) HER2/Neu-Negative (HER2-) Invasive Carcinomas of the Breast. clinicaltrials.gov; 2025. Accessed June 4, 2025. https://clinicaltrials.gov/study/NCT05183828
- Chang CH, Lee YC, Huang CW, Lu YM. Is tamoxifen good enough for the Asian population in ER+ HER2- post-menopausal women with early breast cancer? A nationwide population-based cohort study. PLOS ONE. 2024;19(11):e0313120. doi:10.1371/journal.pone.0313120
- Weiss LK, Burkman RT, Cushing-Haugen KL, et al. Hormone Replacement Therapy Regimens and Breast Cancer Risk. Obstetrics & Gynecology. 2002;100(6):1148.
- Cappello F, Angerilli V, Munari G, et al. FFPE-Based NGS Approaches into Clinical Practice: The Limits of Glory from a Pathologist Viewpoint. J Pers Med. 2022;12(5):750. doi:10.3390/jpm12050750
- Deans ZC, Costa JL, Cree I, et al. Integration of next-generation sequencing in clinical diagnostic molecular pathology laboratories for analysis of solid tumours; an expert opinion on behalf of IQN Path ASBL. Virchows Arch. 2017;470(1):5-20. doi:10.1007/s00428-016-2025-7
Round 2
Reviewer 2 Report
Comments and Suggestions for Authors
- Thank you for carrying out the required modifications!
- What does real-world patients mean in this context?
- Breast and endometrial cancers are classified based on their association with
estrogen signaling. - Still not 100% true about EC. The current WHO classification states the following classification: POLE-ultramutated endometrioid carcinoma, mismatch repair–deficient endometrioid carcinoma, p53-mutant endometrioid carcinoma, and no specific molecular profile (NSMP) endometrioid carcinoma. - Fig 1: Endometrioid patients mean all patients that were included in the endometrioid carcinoma cohort, all endometrium carcinoma/carcinosarcoma patients? Because if the latter, it has to be rephrased. Endometrioid means endomtrium like, and it reflects the morphology of a tumor type, and it is not a synonym for tumors deriving from endometrium! If solely true endometrioid cases are meant here, why is that?
- This comment may be affecting all figures/tables.
- Still do not think the pre-/postmenopausal classification is OK when you are not in possession of the true values...
Author Response
Detailed Response to Reviewers
Thank you for carrying out the required modifications!
1. What does real-world patients mean in this context?
The term “real-world” reflects patients in which data is collected as part of a clinical workflow.
This is as compared to the “research setting”, in which patients are sequenced not due to a need for treatment or clinical action.
In the abstract, the sentence has been revised:
“We analyzed BC and EC, sequenced from patients that received diagnostic tests in oncology clinics….”
2. Breast and endometrial cancers are classified based on their association with
estrogen signaling. - Still not 100% true about EC. The current WHO classification states the following classification: POLE-ultramutated endometrioid carcinoma, mismatch repair–deficient endometrioid carcinoma, p53-mutant endometrioid carcinoma, and no specific molecular profile (NSMP) endometrioid carcinoma.
We thank you for this detailed critique and appreciate the reviewer’s request for these clarifications.
We now see that our opening statements in the Introduction are not aligned with our original intention to introduce the notion that HSD3B1, an enzyme that regulates the production of ligands of the estrogen and progesterone receptors, may be relevant to subsets of breast and endometrial cancer. We have revised the language to reflect the reviewer’s comment.
“Subsets of breast and endometrial cancers express the estrogen receptor and can be driven by receptor signaling activity. Therefore, studying genes and pathways that regulate estrogen or progesterone receptor associated pathways my inform mechanisms of disease pathogenesis. “
Lastly, we recognize that our EC samples are not currently annotated based on WHO molecular classification and this is indicated in the Limitations of the study section:
“Finally, classifications of ECs in this study are based on histology, and therefore the findings may differ if these samples were re-classified based on molecular classification schemes, which are now being adapted in oncology clinics [36]."
3. Fig 1: Endometrioid patients mean all patients that were included in the endometrioid carcinoma cohort, all endometrium carcinoma/carcinosarcoma patients? Because if the latter, it has to be rephrased. Endometrioid means endomtrium like, and it reflects the morphology of a tumor type, and it is not a synonym for tumors deriving from endometrium! If solely true endometrioid cases are meant here, why is that? This comment may be affecting all figures/tables.
No, this is not the case. We sincerely apologize for this continued confusion and spelling errors that led to this.
In Figure 1 and on, the Endometrial Cancer (EC) patients analyzed only include the 3303 that were of the Endometrioid subtype, as indicated in Table 2, which should all be labeled as endometrioid ECs in the revised manuscript.
We have excluded the Carcinosarcoma, Clear cell, and Serous samples from all comparisons, as these subtypes had low sample counts once we stratified tumors by HSD3B1 genotypes, in which the CC group often consistent of few samples. This has been clarified in the Results section:
“As sufficient numbers of each HSD3B1 genotype were only found in the 3303 ECs annotated as the endometrioid subtype (Table 2), the subsequent analyses of ECs focused on examine these tumors. In endometrioid ECs, there was a non-significant trend toward worse median OS in premenopausal patients with the CC genotype compared to those with AC or AA genotypes (Figure 1).”
We also noted multiple areas in the manuscript in which a spelling error likely led to this confusion:
In Table 2 under “ER Positive Prevalence by Subtype” we have revised “Pre-menopausal Endometroid, Post-menopausal Endometroid” to “Pre-menopausal Endometrioid, Post-menopausal Endometrioid”.
Line 237. “These findings indicate that the HSD3B1 genotype and menopausal status together define endometrioid EC with divergent molecular features.”
Line 421. “In endometrioid ECs, we found that independent of menopausal status, tumors harboring the CC genotype generally exhibit reduced oncogenic signaling pathways.”
Line 508. “Based on this cohort, we focused on analyzing endometrioid ECs and excluded other EC Subtypes beyond descriptive demographic analysis due to the limited samples in the pre-menopausal setting (n=4), which limits the statistical power for any analysis.”
Figure 1 “Endometrioid ECs”
Figure 4C “Endometrioid ECs”
Figure 5B, Under Endometrioid ECs, “Pre-menopausal EC, post-menopausal EC”
4. Still do not think the pre-/postmenopausal classification is OK when you are not in possession of the true values...
We agree with this suggestion and have made additions to the “Limitations of the study” section. However, our clinical colleagues have also indicated that menopausal status does impacts clinical decisions on how to treat a patient with breast or endometrial cancer, therefore it is relevant to establish some form of stratification. We therefore have opted to keep an imperfect stratification of the “inferred”menopause status, which is used throughout the revised manuscript.
“Actual menopausal status was not available and therefore key conclusions derived based on the inferred annotation must be further explored through a study with actual menopausal status at the time of cancer diagnosis. Here, menopausal status was approximated using the age at the time of molecular profiling, in which samples from patients older than 55 years were annotated as menopausal. This is based on prior meta-analyses and population studies in BC patients [21,22,34,35]”
Further, in the following key statements we have indicated this is “inferred” menopause status.
Line 32: “with inferred menopausal status assumed by age at molecular profiling”
Line 96 “To comprehensively evaluate the role of HSD3B1 genotypes, our study integrates germline genotype inference with transcriptomic and genomic profiling to analyze its influence on tumor biology and clinical outcomes in BC and EC, stratified by inferred menopausal status and cancer subtype.”
Line 123 “Table 1: Baseline Characteristics of the Breast Cancer Cohort (n = 5085), Including HSD3B1 Genotype, Race, Inferred Menopausal Status, Molecular Subtype, and TMB/MSI Status” and in the table “Inferred Menopausal Status (assumed based on age at molecular profiling)”
Line 126 “Table 2: Baseline Characteristics of the Endometrial Cancer Cohort (n = 5771), Including HSD3B1 Genotype, Race, Inferred Menopausal Status, Molecular Subtype, Estrogen Receptor Status, and TMB/MSI Status.” and in the table “Inferred Menopausal Status (assumed based on age at molecular profiling)”
Line 131 “We evaluated overall survival (OS) across ER+ BC, triple-negative BC (TNBC), and endometrioid EC, stratified by HSD3B1 genotype (AA vs. AC vs. CC) and inferred menopausal status.”
Line 173 “In contrast, CC genotype frequencies among Black and Asian/PI patients were more variable, particularly when further stratified by inferred menopausal status”
Line 197 “To assess whether the HSD3B1 c.1100 genotype is associated with specific somatic alterations, we compared the frequency of key genomic mutations, CNAs, and gene fusions between AA and CC across BC and EC, stratified by inferred menopausal status.”
Line 202 “Further, the differential rates of alterations were dependent on the inferred menopausal status of the patients.”
Line 209, “These findings indicate that inferred menopausal status and the HSD3B1 genotypes are factors that can lead to distinct somatic features, suggestive of the tumors progressing through divergent pathways.”
Line 225, “In endometrioid EC, somatic profiles also differed by genotype and inferred menopausal status.“
Line 236, “These findings indicate that the HSD3B1 genotype and inferred menopausal status together define endometroid EC with divergent molecular features.”
Line 241, “Bar plots display the frequency of select pathogenic mutations, copy number amplification (CNAs), and gene fusions in adrenal-permissive (CC) and adrenal-restrictive (AA) tumors, stratified by cancer type and inferred menopausal status.”
Line 261, “Hormone-related genes, like ESR2 and NR3C2 (encoding the mineralocorticoid receptor), showed no significant differential expression between genotypes across inferred menopausal statuses in TNBC (Figure 4B, and supplementary table 4).”
Line 274, “Volcano plots showing differential gene expression between adrenal-permissive (CC) and adrenal-restrictive (AA) tumors across cancer types and inferred menopausal states. (A) ER+ breast cancer (BC), (B) triple-negative breast cancer (TNBC), and (C) endometrioid endometrial cancer (EC), each stratified by inferred menopausal status.”
Line 297, “In endometrioid EC, CC tumors exhibited broad depletion of transcriptional programs across both inferred menopausal groups.”
Line 314, “Gene set enrichment analysis (GSEA) of hallmark pathways comparing CC vs. AA homozygous tumors, stratified by tumor type and inferred menopausal status”
Line 326, “We analyzed the clinical, genomic, and transcriptomic landscape of HSD3B1 variants across BC and EC subtypes, stratified by inferred menopausal status—representing the largest cohort to date exploring these associations.”
Line 353, “We observed notable differences in HSD3B1 genotype distribution when stratified by cancer subtype and inferred menopausal status.”
Line 414 “HSD3B1 genotypes as a potential regulator of oncogenic signaling and metabolic plasticity across BC subtypes, and that this is dependent on inferred menopause/hormonal status”
Line 421 “we found that independent of inferred menopausal status, tumors harboring the CC genotype generally exhibit reduced oncogenic signaling pathways.
Line 498, Materials and Methods “Inferred menopausal status was assumed based on age at the time of molecular profiling, with <55 years classified as premenopausal and ≥55 years as postmenopausal.
Line 503, “Baseline demographic and clinical characteristics for the BC and EC cohorts—stratified by HSD3B1 genotype, cancer subtype, inferred menopausal status, self-reported race, tumor mutational burden (TMB), and microsatellite instability (MSI)—are summarized in Tables 1 and 2.”
Line 583, “Supplementary Materials: Table 1: HSD3B1 Genotype distributions by inferred menopausal status, per Subtype and Race (Chi-Squared Tests). Table 2: HSD3B1 Genotype distribution by Race as compared to White patients, per Subtype and inferred menopausal Status (Chi-Squared Test).”
We also note that the cohort in which tumor materials have been sequenced based on WTS and WES technology is currently not available for discovery purposes – we perceive this as a strength of this study.
Round 3
Reviewer 2 Report
Comments and Suggestions for Authors
The required modifications have been carried out.